# Region-specific genomic variation and functional divergence of *Helicobacter pylori* clinical isolates from the gastric antrum and corpus

Chih-Ho Lai,[1,2,3,4,5,6] Shih-Yu Wu,[1,7] Shiao-Wen Li,[8] Quan N. U. Ho,[1] Ruei-Lin Chiang,[9] Ngoc-Niem Bui,[1,10] Hui-Yu Wu,[1] Yu-Tsen Huang,[1] Cheng-Hsun Chiu,[1,2] Wen-Ching Wang[11]

**ABSTRACT**  The genetic variability of *Helicobacter pylori* contributes to differences in the severity of gastrointestinal diseases. Within the stomach, *H. pylori* exhibits diverse strain patterns and genetic variations that enable the evolution of new virulence factors, development of antibiotic resistance, and evasion of the host immune system. However, the comprehensive analysis of whole-genome sequences and their functional impact on gastric epithelial cells remains limited. In this study, we performed whole-genome sequencing, *de novo* assembly, and comparative genomic analysis on two pairs of *H. pylori* strains (v225/v226 and v290/v291) isolated from the antrum and corpus of two peptic ulcer patients. Bioinformatic tools were used to annotate and compare genes related to adhesion and virulence. Functional assays were conducted to assess strain-specific pathogenic effects on gastric epithelial cells. The analyses revealed substantial genetic heterogeneity between antral and corpus isolates, particularly in adhesion-related genes (*sabA*, *babA/B*), the cytotoxin-associated gene pathogenicity island (*cag*-PAI) cluster, and *vacA* sequences. Functional assays demonstrated region-specific differences, with corpus strains showing stronger adhesion and pro-inflammatory responses, whereas antral strains exhibited higher vacuolating activity. These findings demonstrate the ability of *H. pylori* to colonize specific stomach regions, undergo genetic diversification, and evolve niche-specific adaptations and pathogenicity in different gastric environments.

**IMPORTANCE**  *Helicobacter pylori* is a major cause of severe gastrointestinal diseases. It can establish persistent colonization in different regions of the stomach, where distinct environmental conditions drive niche-specific adaptation. Here, we found that *H. pylori* evolves genetic diversity in various factors, including virulence factors, adhesion molecules, and outer membrane proteins, to facilitate persistent colonization. Understanding how *H. pylori* generates genetic diversity to support colonization is crucial for developing more effective infection management strategies, improving molecular detection, and refining personalized treatment approaches.

**KEYWORDS**  *Helicobacter pylori*, whole-genome sequence, virulence factor, heterogenicity, pathogenicity, functional divergence

*H*elicobacter pylori is a microaerophilic, spiral-shaped, gram-negative bacterium that colonizes the stomach. The presence of *H. pylori* is closely linked to an elevated risk of developing peptic ulcer disease and gastric adenocarcinoma following persistent infection over a long period (1). DNA fingerprinting data have revealed substantial genetic diversity among different *H. pylori* clinical isolates that surpass the diversity of many other studied bacteria (2). This heightened variability arises from high bacterial

Address correspondence to Chih-Ho Lai, chlai@mail.cgu.edu.tw, or Wen-Ching Wang, wcwang@life.nthu.edu.tw.

The authors declare that the research was conducted in the absence of any commercial or financial relationships that could be construed as a potential conflict of interest.

See the funding table on p. 12.

mutation rates and frequent exchanges of genetic material. These events foster extensive allelic diversity and genetic heterogeneity within the human stomach upon infection with multiple *H. pylori* strains (3).

The pathogenicity of *H. pylori* is predominantly attributed to an array of virulence factors (4). Targeting bacterial adhesins and virulence mechanisms is considered a promising strategy to counteract *H. pylori* infection (5). However, the genetic heterogeneity of these factors allows *H. pylori* to generate diverse variants during adhesion and colonization (6). This is evidenced by allelic variations in genes, such as *vacuolating cytotoxin A* (*vacA*), and the presence of non-conserved DNA segments between strains, including *cag*-pathogenicity islands (*cag*-PAI) (7, 8). Consequently, *H. pylori* can express various virulence factors and trigger adaptive mechanisms within the stomach, all of which contribute to the pathogenicity and potential progression of associated diseases (9).

Genomics data of *H. pylori* strains isolated from distinct regions of the stomach, antrum, and corpus have elucidated the complex features of *H. pylori* colonization patterns (10). A previous study revealed that among the analyzed pairs of strains, 27 of 32 exhibited similarities, and the remaining five pairs displayed disparities (11). Further research has demonstrated that *H. pylori* undergoes rapid within-host evolution, driven by mutation, recombination, and frequent coinfection, with transmission occurring predominantly among close contacts (12). Deep sequencing of both population-level and single-colony isolates from the antrum and corpus of individual patients has shown that most infections originate from a single ancestral strain, which subsequently diversifies within the host (13). Collectively, these findings suggest the dynamic, region-specific evolutionary adaptation of *H. pylori* within the gastric environment.

Despite these findings, no comprehensive exploration has addressed the variations in the whole-genome sequences of different *H. pylori* isolates within the same host and their impact on gastric epithelial cells. This study aimed to fill this gap by investigating the whole-genome sequences of two *H. pylori* strain pairs isolated from the gastric antrum and corpus of patients along with their pathogenicity.

## RESULTS

### Comparative whole-genome and virulence gene analyses of clinical *H. pylori* isolates

The previously isolated *H. pylori* strains (v225/v226 and v290/v291) from two distinct gastric regions (antrum and corpus) were found to contain chimerism in the *vacA* gene between the strains isolated from the corpus (v226 and v291) and the antrum (v225 and v226) (14). In the present study, we first investigated the differences in genetic levels and pathogenicity between two pairs of clinical isolates. Then, whole-genome sequencing was performed, followed by the analysis of their virulence genes. The whole genomes of the two pairs of strains were sequenced using next-generation sequencing and compared with the reference strain 26695 using Mauve. Conserved genes between the two v225/v226 and v290/v291 strain pairs were detected. They are listed in their relative order in Fig. 1. We then performed a genome-wide average nucleotide identity (ANI) analysis using FastANI (Version 1.3) for both pairs of isolates: v225 vs. v226 (99.98% ANI) and v290 vs. v291 (99.42% ANI). These results confirm a high degree of genomic identity (>99% ANI), indicating that the paired strains are closely related and likely derived from a common ancestral strain, with divergence driven by niche-specific adaptation. Subsequently, we compared the genomes of the two clinical strains to analyze the differences in virulence factors. As shown in Fig. S1, v290 was the only strain that did not contain the virulence genes *sabA/hopP. sabB/hopO*, α- (1, 2)-fucosyltransferase (*futA*), and undetermined virulence factors were common deletions that appeared in all isolates.

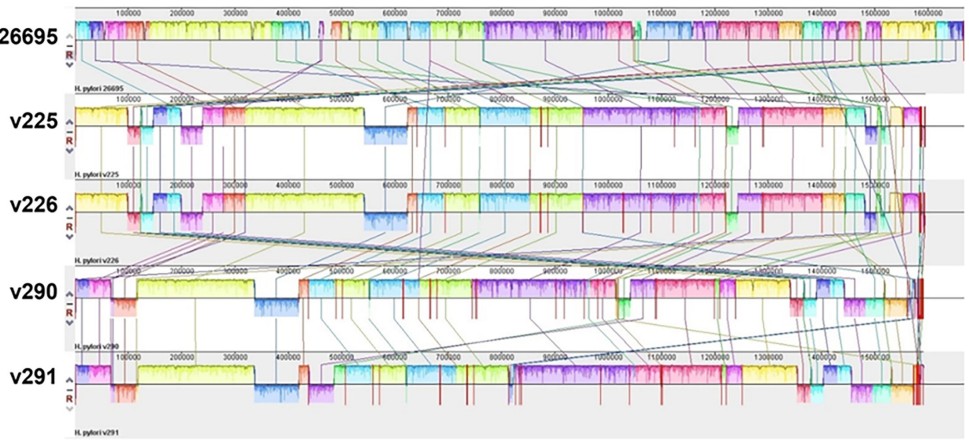

**FIG 1** Genomic divergence between clinical *H. pylori* strain pairs v225/v226 and v290/v291. Each genome is represented horizontally, with homologous segments (locally collinear blocks) indicated by colored lines. Genomic regions inverted relative to the reference strain *H. pylori* 26695 are shown below the central genome line, illustrating structural variations between antral and corpus isolates. Multiple sequence alignments among the various isolates were conducted using Mauve.

## Heterogeneity in virulence genes of *H. pylori* isolates

Although the entire virulence genome of the clinically isolated *H. pylori* strains was not deleted, few differences were detected in the virulence gene sequences between the clinical isolates (Fig. S1). These subtle differences may affect the pathogenicity of *H. pylori* strains. To comprehensively understand the differences in the sequence identity of important virulence factors, we further analyzed multiple sequence alignments between different isolate adhesion genes (*hop* family), urease genes, toxin genes (*cag-PAI* and *vacA*), and immune evasion proteins (*capJ*, *cgat*, etc.). Essential genes for adhesion molecules, such as *hop* family-*sabA* and *babA/B*, were significantly different only in the v290/v291 pair of strains (Table S1). For instance, *sabA* was found only in v291, whereas the similarity between *babA* and *babB* in v290 and v291 was 63 and 59%, respectively. The gene differences of these two strains in *babA/B* were substantial. Moreover, the identity of the virulence genes-*cag-PAI* in the v225/v226 and v290/v291 strains exceeded 90%, and *vacA* accounted for 93 and 89%.

## Variation of adhesion genes in clinical isolates

Successful attachment to gastric epithelial cells is an important step in the initiation of *H. pylori* pathogenicity (15, 16). Adhesion molecules are crucial for *H. pylori* invasion of human gastric epithelial cells (17, 18). Hence, we further analyzed adhesion-related *hop* family genes. Compared to the reference strain 26695, significant differences were evident between *sabA* and *babA/B* in v290/v291. Strain v290 was the only strain with a deletion of the *sabA* gene in its genome. However, no significant differences were determined by the v225/v226 strain pair in the expression of genes related to adhesion (Fig. 2; Fig. S2). Furthermore, both v225 and v226 had a major deletion in *babA/B*. Additionally, in the v290/v291 pair, v291 retained an intact *babA/B* gene, whereas v290 exhibited a large deletion (Fig. 2; Fig. S3 and S4).

## Divergent bacterial adhesion and invasion activities among paired isolates

Concerning the differences in adhesion-related genes between the two pairs of *H. pylori* isolates with slight dissimilarities or large-fragment deletions in the gene sequences of *sabA* and *babA/B*, a previous study found that the balance between SabA and BabA regulations was a crucial factor that assisted *H. pylori* in initiating its colonization and persisting to establish a chronic infection (19). To further explore whether the presence of different adhesion-related genes in the two pairs of isolates (v225/v226 and v290/

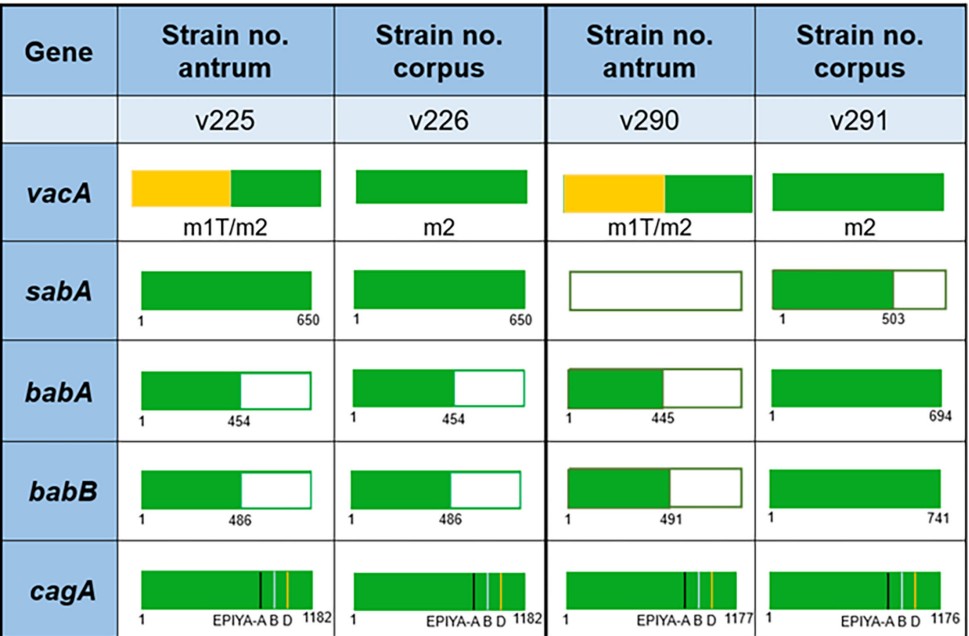

**FIG 2** Comparative analysis of virulence genes reveals region-specific differences in *H. pylori* pathogenesis. Two *H. pylori* strain pairs (v225/v226 and v290/v291) isolated from the antrum and corpus of two peptic ulcer patients were subjected to whole-genome sequencing, *de novo* assembly, and comparative genomic analysis. Adhesion- and virulence-associated genes were annotated and compared, highlighting strain-specific differences in these genes. *cagA*, cytotoxin-associated gene A; *bab*, blood group antigen-binding adhesin; *sab*, sialic acid-binding adhesin; *vac*, vacuolating cytotoxin. Colors represent: green, conserved sequences; yellow, chimeric *vacA* mid-region; open boxes, gene deletions.

v291) affected *H. pylori* pathogenesis, we compared the adhesion and invasion activities of the two pairs of isolates. Adhesion was stronger for the corpus strains (v226 and v291) than for the antrum strains (v225 and v290; Fig. 3A). Additionally, the genetic analysis of adhesion-related genes in strain v291 revealed higher integrity than the adhesion-related genes in strain v290. However, strains v225 and v226 did not differ significantly in their adhesion genes. We then explored the ability of the clinical isolates to invade human gastric epithelial cells. The antral strains v225 and v290 had higher internalization activity than the corpus strains v226 and v291 (Fig. 3B).

## Diversity of the *cag*-PAI gene cluster in *H. pylori* clinical isolates

CagA is one of the most dangerous virulence factors of *H. pylori*, and its toxic activity primarily depends on recognition by the bacterial type IV secretion system (T4SS) (20). The constituent T4SS and *cagA* genes are primarily encoded by *cag*-PAI (21). Accordingly, we analyzed the *cag*-PAI gene cluster in *H. pylori* isolates to determine whether any differences exist in the genes encoding T4SS and CagA. Compared to the reference strain, strains v225 and v226 displayed significantly different identities of the two hypothetical genes (Fig. 4A). In addition, their nearby genes were *cagQ*, *cagP*, *cagC*, and *cagA*. However, in strains v290 and v291, their gene identities were not significantly different (Fig. 4B). Despite the unknown functions of the hypothetical genes, they may affect the expression of adjacent genes. *H. pylori* CagA EPIYA-motif variants exhibit differential activation of host cell signaling pathways, leading to varying contributions to gastric carcinogenesis (22, 23). Thus, we focused on the EPIYA motifs of CagA, which are divided into Western CagA (common type is ABC) and Eastern CagA (common type is ABD). The isolated strains all expressed Eastern-type EPIYA-ABD with no significant difference (Fig. 2; Fig. S5).

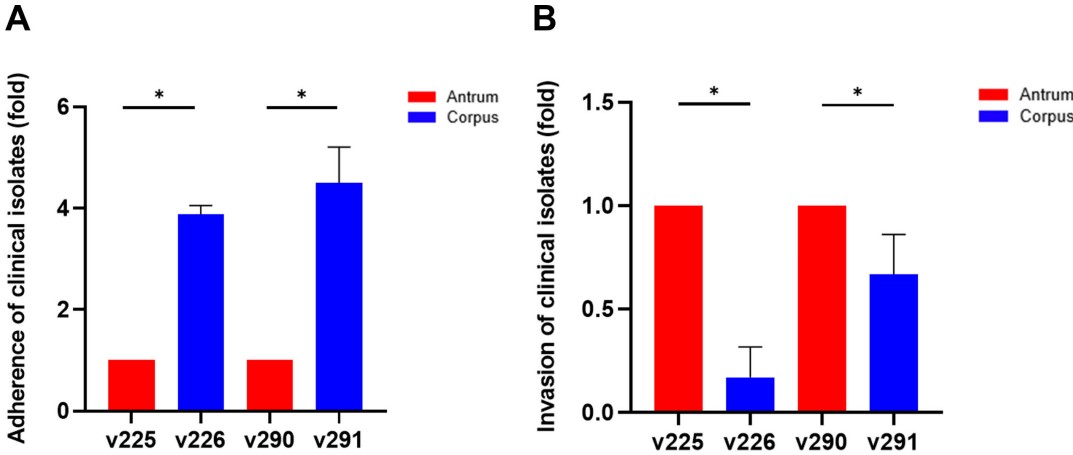

**FIG 3** Comparison of adherence and invasion activities between the antrum and corpus isolates. AGS cells were infected with clinical isolates v225/v226 and v290/v291 at a multiplicity of infection of 100 for 6 h. Following infection, cells were plated on Brucella blood agar plates and incubated for 4–5 days. Colony-forming units were determined to assess the activities of (A) adherence and (B) invasion. Invasion was determined using the gentamicin protection assay. Statistical significance was analyzed using the Student's $t$-test (*, $P < 0.05$).

## Different CagA-induced responses among clinical isolates

The observed differences in the corpus and antrum gastric niches and consequent differences in the adhesion and invasion activities of the isolated strains prompted further examination of the effects of CagA from different strains isolated from human gastric epithelial cells. An analysis of the levels of phosphorylated CagA in the isolates revealed equivalent levels of phosphorylated CagA in strains v225 and v226, while differences were evident in strains v290 and v291, with v291 exhibiting higher phosphorylated CagA levels than v290 (Fig. 5A). To determine whether CagA present in different isolates had a pronounced effect on human gastric epithelial cells, the hummingbird phenotype was used as a biomarker to indicate successful *H. pylori* CagA translocation (25). The v226 and v291 corpus strains displayed a greater ability to induce the hummingbird phenotype compared to the v225 and v290 antrum strains (Fig. 5B through F). Gastric epithelial cells are the first line of defense against pathogens. During pathogenesis, *H. pylori* severely damages cells, and gastric epithelial cells secrete the pro-inflammatory cytokine interleukin-8 (IL-8) (26). Consistent with the percentage of elongated cells, the production of IL-8 was greater in the corpus strains than in the antrum strains (Fig. 5G).

## Heterogeneity of *vacA* from clinical isolates

Differences in the *vacA* sequence result in differences in the ability of cells to generate vacuolization (27, 28). Accordingly, multiple sequence alignments of the v225/v226 and v290/v291 clinical pairs were performed to analyze the differences in *vacA* sequences between the two pairs of isolates. The m-region in *vacA* of strains v225 and v290 isolated from the antrum was considerably different from that of strains v226 and v291 isolated from the corpus. Strains v225 and v290 had the chimeric type (m1T/m2), whereas the m-regions of v226 and v291 were presented in the m2 type (Fig. 2; Fig. S6).

## Diverse *vacA* confers *H. pylori*-induced distinct vacuolation in gastric epithelial cells

We then used an established neutral red assay to examine whether the differences in the *vacA* sequence between the isolated strains had different effects on the vacuolation activity in gastric epithelial cells. Strains v225 and v290 had higher vacuolation activity than v226 and v291 (Fig. 6). The results suggested that the chimeric type (m1T/m2) generated higher toxicity in the antrum-isolated strains (v225 and v290) than in the

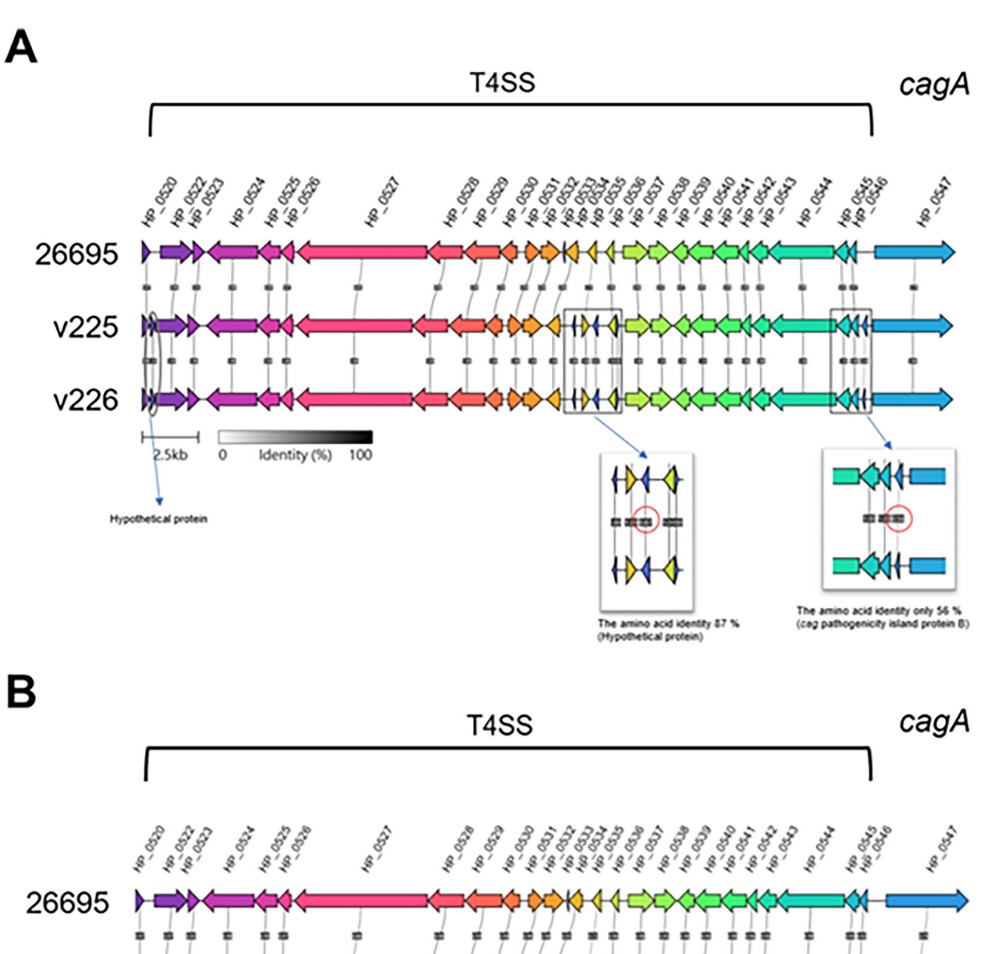

**FIG 4** Variations in the *cag*-PAI gene cluster among *H. pylori* clinical isolates. The *cag*-PAI gene clusters of clinical isolates v225/v226 and v290/v291 were analyzed. Gene clustering was constructed using Clinker (24), with the *cag*-PAI gene cluster of the *H. pylori* reference strain 26695 serving as the baseline for gene orientation. Comparisons were made for clinical strains (A) v225/v226 and (B) v290/v291. The identity of each gene within the *cag*-PAI clusters was expressed as a percentage (%) between gene clusters. Gray circles indicate unaligned genes, while rectangles and red circles highlight genes with relatively large differences in identity.

corpus strains (v226 and v291). These findings demonstrate that *H. pylori* exhibits considerable genetic diversity within the same host, particularly between the antral and corpus regions. This diversity in virulence factors, such as CagA and VacA, contributes to varying levels of pathogenicity in gastric epithelial cells.

## DISCUSSION

To achieve successful pathogenesis, virulence factors of *H. pylori* are instrumental in modulating the host immune system and inducing an inflammatory response (4). *H. pylori* isolates from different gastric regions exhibit nucleotide polymorphisms that enable them to cause severe diseases through specific virulence factors. Our previous study highlighted the genetic variations in *H. pylori* strains isolated from the antrum and corpus of the stomach (14). By using random amplified polymorphic DNA (RAPD) and

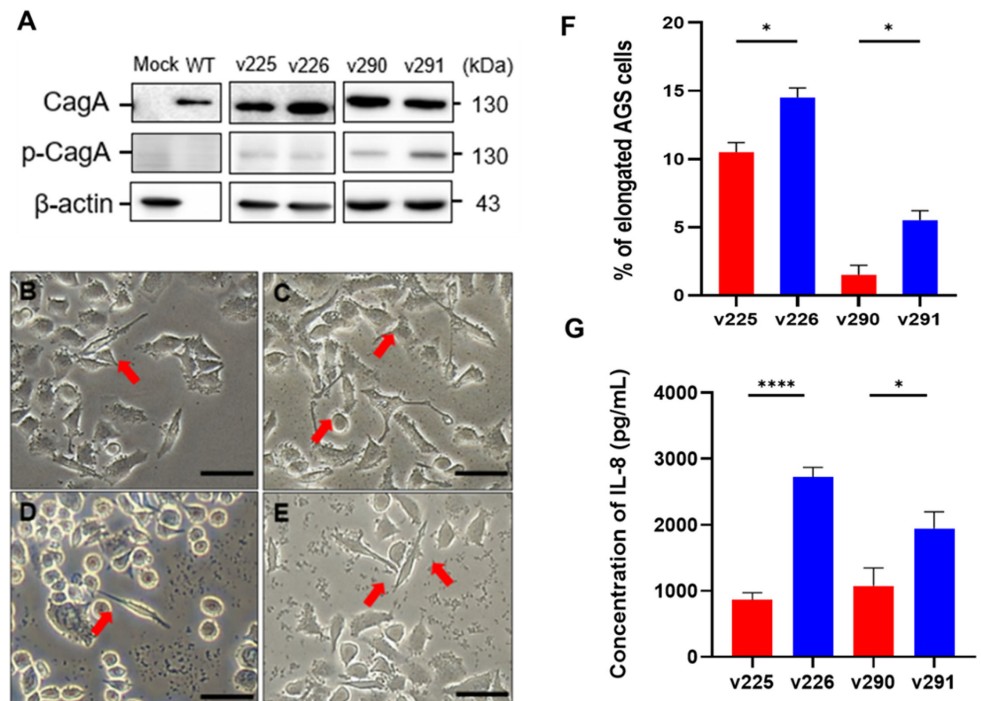

**FIG 5** *H. pylori* CagA-induced different cellular responses by clinical isolates. AGS cells were infected with *H. pylori* 26695 (wildtype, WT) and clinical isolates v225/v226 and v290/v291. (A) CagA translocation and phosphorylation were analyzed using western blotting. Images of the hummingbird phenotype induced by infection with clinical isolates are shown for (B) v225, (C) v226, (D) v290, and (E) v291. Scale bars denote 50 µm. The red arrows indicate cells displaying scattered phenotypes. (F) Quantified percentage of elongated cells (longer than 20 µm). (G) IL-8 levels in the culture supernatant were measured using a standard ELISA method. Statistical significance was assessed using the Student *t*-test (*, $P < 0.05$; ****, $P < 0.001$).

sequence analyses, this study further validated the chimerism in the *vacA* between isolates from these two gastric sites. However, whole-genome sequencing and pathogenicity of these clinical isolates were not thoroughly explored. In the present study, whole-genome sequencing comprehensively identified genetic differences between strain pairs. Our results showed that strains from the antrum and corpus within the same host share a high degree of genomic identity (>99%), supporting the hypothesis of microevolutionary divergence rather than distinct clonal colonization. Our findings reveal significant genetic diversity among *H. pylori* strains from different gastric regions within the same host. This diversity actively contributes to the expression of distinct virulence factors by *H. pylori*, which are pivotal in influencing pathogenic outcomes.

Histological analyses revealed significant differences in gastric antrum and corpus changes between patients with mixed *H. pylori* infections and those infected with a single strain (29). Patients with mixed infections have a notably higher incidence of antral intestinal metaplasia than those with a single infection. Strains of *H. pylori* obtained from a single host exhibit variation in virulence factors and antibiotic resistance patterns (30). Furthermore, the prevalence of mixed *H. pylori* infections is reportedly significantly higher in patients with duodenal ulcers than in those with other gastrointestinal disorders (31). These lines of evidence indicate that mixed *H. pylori* infections may enhance bacterial adaptability to the rigorous gastric environment, contributing to disease progression.

Genomic analyses of *H. pylori* clinical isolates have identified associations between bacterial gene polymorphisms and specific stomach niches (32). These results indicate that chemotaxis, regulatory functions, and outer membrane proteins play crucial roles in *H. pylori* adaptation to the antral and oxyntic mucosa. Recently, deep population and single-colony isolate sequencing revealed the prevalent diversity of *H. pylori* within and

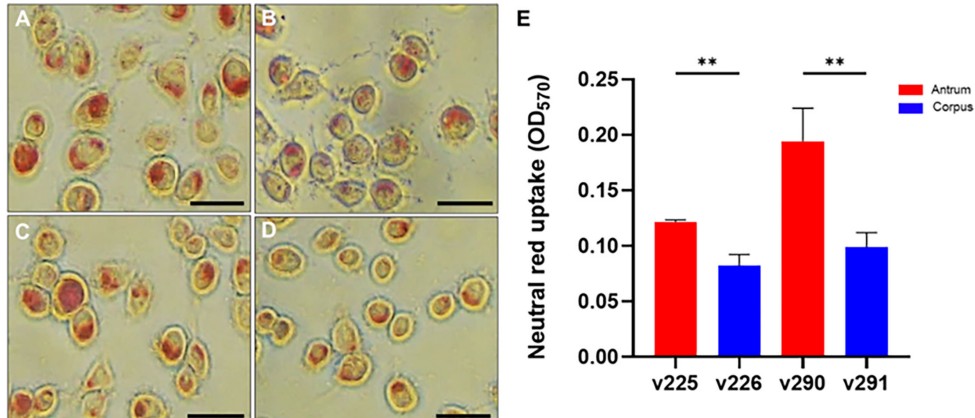

**FIG 6** Distinct vacuolation activity between the antrum and corpus strains. AGS cells were infected with clinical isolates (A) v225, (B) v226, (C) v290, and (D) v291 for 16 h. Vacuoles in the cells were observed using the neutral red uptake assay. Scale bars denote 50 μm. (E) Vacuolation activity was quantified using the neutral red uptake assay. Statistical significance was assessed using the Student's *t*-test (**, $P < 0.01$).

between bacterial populations in the antrum and corpus regions (13). This diversity is closely associated with bacterial virulence and colonization. Collectively, these findings highlight the adaptive mechanisms that enable *H. pylori* to colonize distinct gastric environments across multiple isolates from different patients.

It is increasingly evident that various virulence factors produced by *H. pylori* can facilitate establishment within the host niche (6, 33). Our recent findings have highlighted how these virulence factors activate host innate sensing mechanisms, thereby promoting inflammation in gastric epithelial cells (34, 35). Genomic variability among *H. pylori* strains is closely linked to disease occurrence and progression mainly through genetic variations in virulence factors (36). Differences in the pathogenicity of these virulence factors, particularly those containing *cagA* and *vacA s1m1*, are associated with a high risk of disease development (37, 38). Genetic variations in virulence factors have been observed in different parts of the stomach (30, 39). The host immune response can drive genetic heterogeneity in *H. pylori*, with strains colonizing different stomach regions facing distinct selection pressures, such as variations in pH levels or immune defense. These conditions cause adaptive mutations or distinct genetic rearrangements, making bacteria more suited to harsh environments (40).

Regarding the adhesion factors *sabA/B* and *babA/B*, a previous study reported that *babA* evolves adaptively to enhance adhesion in response to more acidic environments (41). Although sequence differences in *babA* were observed only in the v290/v291 pair in our genetic analysis, strains v226 and v291 isolated from the more acidic region of the stomach (corpus) exhibited higher adhesion activity in *in vitro* experiments. Our hypothesis that this enhanced adhesion may assist bacteria in better hiding under the gastric mucosa is supported by earlier findings that adhesion factors bind to various antigens in the human stomach. This binding plays a crucial role in persistent infection and enhances CagA translocation via T4SS, leading to gastric epithelial cell damage (42).

CagA exhibits mutual antagonism with VacA (43, 44), although the exact mechanism underlying this functional antagonism remains unclear. Cells rarely exhibit both vacuolation and hummingbird phenotypes simultaneously because VacA and CagA downregulate each other's effects on gastric epithelial cells (45). In the present study, compared to the corpus strains (v226 and v291), the antrum strains (v225 and v290), which exhibited stronger vacuolating activity, showed weaker CagA-induced morphological changes and reduced IL-8 release, providing evidence for the mutual antagonism between VacA and CagA. Additionally, v226 and v291, which had stronger adhesion activities, demonstrated higher CagA translocation and phosphorylation activities, confirming that *H. pylori* adhesion contributes to CagA activity. Although functional

antagonism between CagA and VacA has been observed in the present study, the underlying molecular mechanisms remain to be fully elucidated and merit further investigation.

## Conclusions

Our study demonstrates that *H. pylori* strains isolated from the antrum and corpus of the same patient exhibit marked genetic heterogeneity, particularly in adhesion-related genes and key virulence factors, such as *cagA*, *vacA*, and *cag*-PAI. WGS and comparative analyses revealed region-specific genotypic differences, which translated into distinct functional outcomes in gastric epithelial cells. Corpus-derived strains showed enhanced adhesion, CagA translocation and phosphorylation, and IL-8 induction, indicating a more pro-inflammatory profile, whereas antral strains induced greater VacA-mediated vacuolation. These findings emphasize the importance of considering intra-host *H. pylori* diversity in pathogenesis and suggest that spatial variation in strain characteristics may influence disease progression and treatment strategies.

## MATERIALS AND METHODS

### *H. pylori* strains

Clinical strains v225/v226 and v290/291 were isolated from the gastric antrum and corpus of two patients (nos. 20A and 35A), respectively (14). Both patients were diagnosed with peptic ulcers based on endoscopy findings. The strains isolated from the antrum (v225 and v226 and v290 and v291) were analyzed using distinct restriction fragment length polymorphism patterns and random amplified polymorphic DNA. *H. pylori* 26695 (ATCC 700392) was used as a reference strain for whole-genome sequence comparisons. All *H. pylori* strains were cultured on Brucella agar plates (Becton Dickinson, USA) supplemented with 10% sheep blood.

### Preparation of *H. pylori* genomic DNA

Bacterial genomic DNA was extracted using a WelPrep DNA kit following the manufacturer's protocol (Cat. No. D001; Welgene Biotech, Taiwan). DNA samples with an optical density ratio of 260/280 nm ranging from 1.8 to 2.0 and a Qubit vs NanoDrop quantity ratio > 0.7 were selected for subsequent processing.

### Library construction

In total, 10 µg of DNA was sonicated using a Misonix 3000 sonicator, resulting in fragment sizes ranging from 400 to 500 base pairs (bp). The size distribution of the DNA fragments was validated using a bioanalyzer equipped with a DNA 1000 chip (Agilent Technologies, USA). One microgram of the sonicated DNA underwent end-repair, A-tailing, and adaptor ligation according to the Illumina TruSeq DNA preparation protocol.

### Genome *de novo* assembly

Each sample comprised paired-end reads with a length of approximately ±300 bp. Quality control, trimming, and filtering procedures were performed utilizing BBDuk (https://jgi.doe.gov/data-and-tools/bbtools/bb-tools-user-guide/bbduk-guide/). Subsequently, the cleaned and filtered nuclear reads were subjected to *de novo* assembly using Spades 3.15.1 with default settings (46).

### Gene annotation

Open reading frames were annotated using the rapid annotation with subsystem technology (47). The generated predictions were cross-referenced against the National

Center for Biotechnology Information (NCBI)'s non-redundant (nr) database using BLASTp.

## Multiple sequence alignments and comparison of analyzed genomes

Data for comparative analysis were obtained from the NCBI database encompassing the complete sequences and annotations of *H. pylori* isolates. Multiple sequence alignments among the various isolates were conducted using Mauve (48). Amino acid sequences of virulence genes were aligned with the reference genome *H. pylori* 26695 using BLASTp, and the gene cluster was constructed using Clinker (24). Genome-wide average nucleotide identity (ANI) analyses for both isolate pairs were conducted using FastANI (version 1.3) (49).

## Cell culture

Human gastric adenocarcinoma AGS cells (ATCC CRL 1739) were maintained in F12 medium (Sigma-Aldrich, USA) supplemented with 10% heat-inactivated fetal bovine serum (FBS) (HyClone, USA) at 37°C in a 5% $CO_2$ for 24–48 h.

## *H. pylori* adhesion and invasion on AGS cells

Adhesion and invasion of *H. pylori* to AGS cells were assessed using a standard colony formation assay, as described previously (34). Briefly, AGS cells ($2 \times 10^5$) were seeded in 12-well plates containing F12 medium supplemented with 10% FBS and incubated at 37°C. Following incubation, cells were infected with *H. pylori* at a multiplicity of infection (MOI) of 100 for 6 h. After infection, unattached *H. pylori* were removed by washing the cells twice with PBS. Subsequently, the *H. pylori*-infected AGS cells were lysed with 1 mL of sterilized water for 10 min. The lysates were serially diluted in PBS, plated on Brucella blood agar plates, and incubated for 4–5 days. Viable colony-forming units (CFUs) were then counted. To determine the number of bacteria invading viable cells, a gentamicin protection assay was employed (50). AGS cells were infected with *H. pylori* for 6 h, and then treated with 100 µg/mL of gentamicin (Sigma-Aldrich, USA) for 1.5 h at 37°C to remove extracellular bacteria. Following treatment, the cells were processed using the same procedure described above to determine the number of viable CFUs.

## Analysis of *H. pylori vacA*-induced cellular vacuolation

Cellular vacuolation induced by *H. pylori vacA* was evaluated using a neutral red uptake assay (51). AGS cells ($2 \times 10^5$) were seeded in 24-well plates and incubated at 37°C for 16 h. Subsequently, the cells were infected with *H. pylori* at a MOI of 100 for 12 h. After infection, the cells were washed with PBS and incubated with 0.05% neutral red (Sigma-Aldrich, USA) for 4 min. Acidified alcohol (1% 12 N HCl in 75% ethanol) was then added to elute the neutral red, which was measured at OD 570 nm using a spectrophotometer (Molecular Devices, USA).

## Analysis of the hummingbird phenotype

AGS cells ($0.5 \times 10^5$) were seeded in 24-well plates containing F12 medium supplemented with 10% FBS and incubated at 37°C. Following incubation, cells were infected with *H. pylori* at a MOI of 100 for 8 h. Cells exhibiting an elongated morphology (hummingbird phenotype) were identified as those with needle-like protrusions longer than 20 µm, as previously described (52). The number of cells displaying the hummingbird phenotype was counted among one hundred AGS cells, and the proportion was calculated. Each sample was examined in triplicate across at least three independent experiments.

## Measurement of IL-8 production

AGS cells ($2 \times 10^5$) were seeded in 24-well plates containing F12 medium supplemented with 10% FBS and maintained at 37°C. Following incubation, cells were infected with *H. pylori* at the MOI of 100 for 8 h. The supernatant from the cell culture was collected, and the level of IL-8 was quantified by sandwich enzyme-linked immunosorbent assay (ELISA) following the manufacturer's instructions (R&D Systems, USA).

## Analysis of CagA translocation and phosphorylation

The assessment of CagA translocation and phosphorylation levels was performed by following the previous study (50). Briefly, AGS cells were infected with *H. pylori* at the MOI of 100 for 6 h. Subsequently, cell lysates were prepared and subjected to 6.5% SDS-PAGE, followed by transfer onto polyvinylidene difluoride (PVDF) membranes for western blot assay. CagA and phospho-CagA were probed using a mouse anti-CagA antibody (Santa Cruz Biotechnology, USA) and an anti-phosphotyrosine antibody (4G10) (Millipore, USA), respectively. Proteins of interest were visualized using the enhanced chemiluminescence reagent (GE Healthcare, USA) and analyzed by AzureSpot analysis software with Azure 400 (Azure Biosystems, USA).

## Statistical analysis

The experimental data were presented as means ± standard error of the mean. Student's *t*-test was employed to assess the significance of differences between the two groups. The difference was considered statistically significant when $P < 0.05$. Statistical analyses were conducted using Prism 8.0

## ACKNOWLEDGMENTS

The authors sincerely appreciate assistance in analyzing the *H. pylori* genomic sequences and bioinformatic analysis from the Next Generation Sequencing Core, Bioinformatics Core, and Molecular Medicine Research Center, Chang Gung University, Taoyuan, Taiwan.

This study was supported by the National Science and Technology Council (112-2320-B-182-036-MY3, 112-2320-B-182-042-MY3, and 114-2321-B-182A-002 to Chih-Ho Lai; 112-2320-B-007-004 -MY3 and 114-2320-B-007-001 to Wen-Ching Wang), Chang Gung Memorial Hospital (CMRPD1L0321, CMRPD1M0491-2, CORPD1M0021-3, and CMRPD1P0161), Research Center for Emerging Viral Infections from the Featured Areas Research Center Program within the framework of the Higher Education Sprout Project by the Taiwan Ministry of Education, and Tomorrow Medical Foundation.

C.H.L.: conceived the study, wrote the manuscript, and reviewed the final version of this manuscript. S.Y.W.: performed the experiments, analyzed the data, and wrote the manuscript. S.W.L.: performed the experiments, analyzed the data, and wrote the manuscript. Q.N.U.H.: analyzed the data and wrote the manuscript. R.L.C.: performed the experiments and analyzed the data. N.N.B.: analyzed the data and wrote the manuscript. H.Y.W.: performed the experiments and analyzed the data. Y.T.H.: performed the experiments and analyzed the data. C.H.C.: conceived and designed the experiments. W.C.W.: conceived the study, wrote the manuscript, and reviewed the final version of this manuscript. All authors reviewed and approved the manuscript.

## AUTHOR AFFILIATIONS

[1]Department of Microbiology and Immunology, Graduate Institute of Biomedical Sciences, Chang Gung University, Taoyuan, Taiwan

[2]Molecular Infectious Disease Research Center, Department of Pediatrics, Chang Gung Memorial Hospital at Linkou, Taoyuan, Taiwan

[3]Department of Microbiology and Immunology, China Medical University, Taichung, Taiwan

[4]Department of Nursing, Asia University, Taichung, Taiwan

[5]Center for Molecular and Clinical Immunology, Institute of Immunology and Translational Medicine, Chang Gung University, Taoyuan, Taiwan

[6]Research Center for Emerging Viral Infections, Chang Gung University, Taoyuan, Taiwan

[7]Department of Laboratory Medicine, Kaohsiung Chang Gung Memorial Hospital, Kaohsiung, Taiwan

[8]Department of Life Sciences, National University of Kaohsiung, Kaohsiung, Taiwan

[9]Molecular Medicine Research Center, Chang Gung University, Taoyuan, Taiwan

[10]Department of Microbiology, Can Tho University of Medicine and Pharmacy, Can Tho, Vietnam

[11]Department of Life Science, Institute of Molecular and Cellular Biology, National Tsing Hua University, Hsinchu, Taiwan

## AUTHOR ORCIDs

Chih-Ho Lai  http://orcid.org/0000-0001-7145-784X

Wen-Ching Wang  http://orcid.org/0000-0002-7422-3667

## FUNDING

| Funder | Grant(s) | Author(s) |
| --- | --- | --- |
| National Science and Technology Council | 112-2320-B-182-036-MY3, 112-2320-B-182-042-MY3, and 114-2321-B-182A-002 | Chih-Ho Lai |
| Chang Gung Memorial Hospital, Linkou | CMRPD1L0321, CMRPD1M0491-2, CORPD1M0021-3, and CMRPD1P0161 | Chih-Ho Lai |
| National Science and Technology Council | 112-2320-B-007-004-MY3 and 114-2320-B-007-001 | Wen-Ching Wang |

## AUTHOR CONTRIBUTIONS

Chih-Ho Lai, Conceptualization, Writing – review and editing | Shih-Yu Wu, Data curation, Investigation, Writing – original draft | Shiao-Wen Li, Data curation, Investigation, Writing – original draft | Quan N. U. Ho, Data curation, Writing – original draft | Ruei-Lin Chiang, Data curation, Investigation | Ngoc-Niem Bui, Data curation, Writing – original draft | Hui-Yu Wu, Data curation, Investigation | Yu-Tsen Huang, Data curation, Investigation | Cheng-Hsun Chiu, Conceptualization, Supervision | Wen-Ching Wang, Conceptualization, Writing – review and editing

## DATA AVAILABILITY

The genomic sequence files supporting the findings of this article are available in the NCBI Sequence Read Archive under BioProject (https://www.ncbi.nlm.nih.gov/bioproject/PRJNA1221638). All relevant data are contained within the article. The original contributions presented in the study were included in the main article and supplementary material. Further inquiries can be directed to the corresponding authors.

## ETHICS APPROVAL

This study was approved by the Institutional Review Board of Chang Gung Memorial Hospital, Linkou, Taiwan (IRB nos. 201701000A3 and 202000374A3). All patients signed informed consent and participated in the study.

## ADDITIONAL FILES

The following material is available online.

## Supplemental Material

**Supplemental material (mSystems01029-25-s0001.pdf).** Supplemental figures and table.

## Open Peer Review

**PEER REVIEW HISTORY (review-history.pdf).** An accounting of the reviewer comments and feedback.

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
