## [Reviewer comments · mSystems]

Region-specific genomic variation and functional divergence of *Helicobacter pylori* clinical isolates from the gastric antrum and corpus

Chih-Ho Lai, Shih-Yu Wu, Shiao-Wen Li, Quan N. U. Ho, Ruei-Lin Chiang, Ngoc-Niem Bui, Hui-Yu Wu, Yu-Tsen Huang, Cheng-Hsun Chiu, and Wen-Ching Wang

Corresponding Author(s): Chih-Ho Lai, Chang Gung University

Review Timeline:

Submission Date:

July 10, 2025

Accepted:

August 22, 2025

Editor: Saheed Imam

Reviewer(s): The reviewers have opted to remain anonymous.

Transaction Report:

DOI: <https://doi.org/10.1128/msystems.01029-25>

Re: mSystems01029-25 (**Region-specific genomic variation and functional divergence of *Helicobacter pylori* clinical isolates from the gastric antrum and corpus**)

Dear Dr. Chih-Ho Lai:

Your manuscript has been accepted, and I am forwarding it to the ASM production staff for publication. Your paper will first be checked to make sure all elements meet the technical requirements. ASM staff will contact you if anything needs to be revised before copyediting and production can begin. Otherwise, you will be notified when your proofs are ready to be viewed.

Sincerely,
Saheed Imam
Editor
mSystems

Reviewer #1 (Comments for the Author):

The authors addressed all the reviewer's comments from previous review by conducting genome-wide average nucleotide identity (ANI) analysis, incorporating recent references, revising the Discussion section to clarify the distinction between strain-level variation and microevolution, modifying figures, and correcting some typos.